# Effects of Low Light after Heading on the Yield of Direct Seeding Rice and Its Physiological Response Mechanism

**DOI:** 10.3390/plants12244077

**Published:** 2023-12-06

**Authors:** Peng Ma, Lin Zhou, Xue-Huan Liao, Ke-Yuan Zhang, Li-Se Aer, Er-Luo Yang, Jun Deng, Rong-Ping Zhang

**Affiliations:** School of Life Science and Engineering, Southwest University of Science and Technology, Mianyang 621010, China; mapeng7815640@163.com (P.M.); zhoulin@mails.swust.edu.cn (L.Z.); lxh000818@163.com (X.-H.L.); ky010302@163.com (K.-Y.Z.); rm88886666@163.com (L.-S.A.); 18882457656@163.com (E.-L.Y.); 15528510375@163.com (J.D.)

**Keywords:** shading, direct seeding rice, chlorophyll, enzyme activity, production

## Abstract

As a photophilous plant, rice is susceptible to low-light stress during its growth. The Sichuan Basin is a typical low-light rice-producing area. In this study, eight rice varieties with different shade tolerances were studied from 2021 to 2022. The physiological adaptability and yield formation characteristics of rice were studied with respect to photosynthetic physiological characteristics and dry matter accumulation characteristics, and the response mechanism of rice to low light stress was revealed. The results showed that the shading treatment significantly increased the chlorophyll a, chlorophyll b, and total chlorophyll contents in the leaves of direct-seeded rice after heading, and the total chlorophyll content increased by 1.68–29.70%. Nitrate reductase (NR) activity first increased and then decreased under each treatment, and the shading treatment reduced the NR activity of direct-seeded rice. Compared to the control treatment, the peroxidase (POD) activity of each variety increased from 7 to 24 d after the shading treatment. The transketolase (TK) activity in direct-seeded hybrid rice increased under low light stress. Compared with the control, shading treatment significantly reduced the aboveground dry matter, grain number per panicle, and seed setting rate of direct-seeded rice at the full heading stage and maturity stage, thus reducing the yield of direct-seeded rice by 26.10–34.11%. However, under the shading treatment, Zhenliangyou 2018 and Jingliangyou 534 maintained higher chlorophyll content and related enzyme activities, accumulated more photosynthetic products, and reduced yield. In general, Zhenliangyou 2018 and Jingliangyou 534 still had a yield of 7.06–8.33 t·hm^−2^ under low light. It indicated that Zhenliangyou 2018 and Jingliangyou 534 had better stability and stronger tolerance to weak light stress and had a higher yield potential in weak light areas such as Sichuan.

## 1. Introduction

Rice is one of the most important food crops in the world and provides a staple food for over half of the world’s population [1,2]. Rice production in China is the highest in the world, accounting for 2/5 of the world’s total rice production, and is a staple food for over 60% of China’s population [3]. However, with the development of cities and population growth, the serious shortage of rural labor resources, environmental pollution, water shortages, and other issues have become increasingly prominent, and rice production improvement is facing great challenges. Light is an important factor in rice production. Changes in light intensity and quality directly lead to changes in the physiological indicators, yield, and quality of rice [4,5]. In recent years, ‘shading’ caused by air pollution and clouds has become an important limiting factor for rice production in China, India, and other countries [6]. Severe haze and aerosol pollution may further reduce solar irradiance by 28–49% [7]. Sichuan is the largest industrial province in Southwest China, and there is a serious risk of air pollution. In addition, the Sichuan Basin is perennially cloudy, with annual sunshine hours of 1200 h and annual solar radiation of 3345–3763 MJ m^−2^, which is a typical low-light rice area in China [8,9]. Weak light severely restricted rice production in Sichuan. Therefore, it is important to study the physiological indices of rice plants under low-light conditions for rice production in Sichuan. Superoxide dismutase (SOD), peroxidase (POD), and catalase (CAT) are the key enzymes involved in scavenging reactive oxygen species. Malondialdehyde (MDA) is a membrane lipid peroxidation product, and its content is a symbol of plant health, reflecting the degree of cell membrane damage under stress conditions; the higher the MDA content, the more serious the cell membrane damage [10,11]. Under low-light stress, the high POD enzyme activity causes shade-tolerant rice to exhibit strong resistance [12]. With the increase in low-light stress intensity and the extension of time, the increase in MDA content in rice was greater, and damage to the membrane system was more serious [13]. Additionally, dynamic changes in carbon and nitrogen metabolism directly affect the formation, transport, and protein synthesis of rice photosynthetic products, which largely determine the yield and quality of rice [14]. Nitrate reductase (NR) activity, which is closely related to nitrogen metabolism, was significantly lower under low light than under natural light. After light returns to normal levels, NR activity returns to normal levels [13]. Transketolase (TK) plays a central role in the Calvin cycle. When the TK gene (TKL) is inhibited in tobacco, the photosynthesis rate decreases with decreased TK activity [15]. Some studies have also found that TK does not limit photosynthesis because an increase in TK activity by 76–150% does not enhance the photosynthesis of tobacco [16], and an increase in its content does not enhance the photosynthesis of rice [17]. Chlorophyll, a photosynthetic pigment, also showed significant differences under low-light conditions. Low light intensity increases the chlorophyll a, chlorophyll b, and total chlorophyll contents in the leaves of rice at the filling stage and decreases the value of chlorophyll a/b and the net photosynthetic rate [18,19,20]. This disturbs dry matter accumulation, transport, and distribution in rice and reduces the number of effective panicles and the seed setting rate, resulting in a decrease in yield [21,22,23]. Varieties with stronger shade tolerance had higher light-use efficiency and less reduction in dry matter accumulation and yield; however, the effects of shading stress on rice vary with variety and cultivation method. The effects of shade stress on plant physiological indices, dry matter accumulation, and the yield of direct-seeded rice are unclear. Therefore, in this study, eight hybrid rice varieties were used to conduct a two-year artificial direct seeding and shading test in Mianyang, Sichuan, to study the physiological characteristics, dry matter accumulation, and yield performance of their plants and to provide a theoretical basis for selecting direct seeding rice varieties after wheat and formulating cultivation measures in Sichuan and other low-light areas.

## 2. Results

### 2.1. Effects of Shading on the Chlorophyll Content of Direct-Seeded Rice 

The shading treatment significantly delayed the decreasing trend of chlorophyll a, chlorophyll b, and total chlorophyll contents in the leaves of direct-seeded rice after heading (Figure 1, Figure 2, Figure 3 and Figure 4). Under shading treatment, except for the chlorophyll a and total chlorophyll content of C Liangyou 0861, which showed a trend of decreasing first, then increasing, and then decreasing under shading conditions, the chlorophyll a, chlorophyll b, and total chlorophyll content of other varieties showed a trend of increasing first and then decreasing. In addition, the values of chlorophyll a, chlorophyll b, and total chlorophyll of most varieties under shading treatment for 14 days (S-14) and 21 days (S-21) were the same as those of the control for 7 days (S-7) and 14 days (S-14). Under the control treatment, the chlorophyll a, b, and total chlorophyll contents of Yangxianyou 919 and Liangyouhuazhan showed a continuous downward trend. The total chlorophyll content of direct-seeded rice under shading treatment was 1.68–29.70% higher than that of the control. On the 28th day of shading treatment, the incremental order of chlorophyll a content in direct seeding rice leaves compared with the control was as follows: Huiliangyou Yuehesimiao > Shenzhenliangyou 2018 > C Liangyou 0861 > Jingliangyou 534 > Yangxianyou 919 > C Liangyou Huazhan > Jiuyou 27 zhan > Longliangyou Yuehesimiao. The order of chlorophyll b content compared with the control was: Shenzhenliangyou 2018 > Huiliangyou Yuehesimiao > C Liangyou 0861 > Jingliangyou 534 > Yangxianyou 919 > C Liangyou Huazhan > Jiuyou 27 zhan > Longliangyou Yuehesimiao. At the same time, the changes in the chlorophyll a/b values in the leaves of each direct-seeded rice were not the same. Under the shading treatment, the chlorophyll a/b value of Jingliangyou 534 first increased and then decreased, Liangyou Huazhan decreased, and the rest decreased gradually. Under the control treatment, except for Zhenliangyou 2018 and Longliangyou Yuehesimiao, the chlorophyll a/b values of the varieties first increased and then decreased. The chlorophyll a/b value of Jingliangyou 534 increased after 7 d of shading treatment (S-7). From 14 d of shading treatment (S-14) to 28 d of shading treatment (S-28), with the passage of shading treatment time, the chlorophyll a/b values of each variety decreased to varying degrees under shading treatment, suggesting that rice plants can gradually adapt to stressful environments by adjusting the chlorophyll components in the leaves.

### 2.2. Effects of Shading on NR, POD, and Other Enzyme Activity in Directly Seeded Rice 

NR is a key enzyme in nitrogen assimilation in plants. The results of the two-year experiment showed that the shading treatment significantly reduced NR enzyme activity (Figure 5 and Figure 6). From the test results in 2021, under the shading treatment, C Liangyou 0861, Jingliangyou 534, Yangxianyou 919, and Longliangyou Yuehesimiao exhibited a decreasing trend, followed by an increase and then another decrease. Meanwhile, Zhenliangyou 2018, Jiuyou 27 zhan, Huiliangyou Yuehesimiao, and C Liangyou Huazhan gradually decreased with time. Except for Jiuyou 27 and Huiliangyou Yuehesimiao, which reached maximum activity after 7 d of shading (S-7), the rest reached maximum activity after 14 d of shading (S-14). Under the control treatment, the NR enzyme activity of each variety first increased and then decreased. From the 2022 results, under shading treatment, the NR enzyme activity of Zhenliangyou 2018 and C Liangyou Huazhan gradually decreased with time; the NR enzyme activity of Huiliangyou Yuehe Simiao showed a trend of decrease-increase-decrease; and the NR enzyme activity of C Liangyou 0861, Jingliangyou 534, and other five varieties increased first and then decreased. Except for Zhenliangyou (2018), Jingliangyou (534), Yangxianyou (919), Jiuyou 27 zhan, and C Liangyouhuazhan reached their maximum value at 7 d of shading treatment (S-7); the rest reached their maximum value after 14 d of shading treatment (S-14). Under the control treatment, the NR activity of each variety showed an increasing and then decreasing trend.

As shown in Figure 7, under shading treatment, except for C Liangyou 0861, the POD activity of 7 varieties such as Zhenliangyou 2018 showed a trend of increasing first and then decreasing, and the POD activity of Jiuyou 27 zhan, Huiliangyou Yuehesimiao, and C Liangyouhuazhan reached the highest at 14 d of shading (S-14), and the other 5 varieties reached the highest at 7 d of shading (S-7). Under the control treatment, except for C Liangyou 0861, the trend of change in the POD activity of the other varieties was consistent with that of the shading treatment. Compared to the control treatment, the POD activity of each directly seeded rice leaf significantly increased from 7 d (S-7) to 24 d (S-24). Shading treatment for 21–28 d (S-28) reduced the POD activity in the leaves of Zhenliangyou 2018, whereas the POD activity of the remaining varieties increased. 

The higher the MDA content, the greater the degree of crop damage. The shading treatment significantly increased the MDA content in the leaves of direct-seeded rice (Figure 8). The MDA content in different rice varieties was different under the shading treatment but showed a trend of continuous increase, and the MDA content under the control treatment also showed an upward trend. In addition, the increase in MDA content in Yangxianyou 919 and C Liangyouhuazhan under shading and control treatments was smaller than that in the other varieties, and the effects of shading treatment on TK enzyme activity in different varieties of direct-seeding rice after heading were not the same (Figure 9). After shading treatment for 7 d (S-7), the TK enzyme activity of Zhenliangyou 2018, Jiuyou 27 zhan, and Huiliangyou Yuehesimiao decreased by 1.18–7.81%; shading for 14 d (S-14) reduced the TK enzyme activity of Jingliangyou 534 and Huiliangyou Yuehesimiao by 7.42–15.05%. Shading for 21 d (S-21) reduced only the TK enzyme activity of Huiliangyou Yuehesimiao. After 28 d of shading treatment (S-28), the TK enzyme activity of each variety increased by 3.89–20.49%. Compared with the control, Zhenliangyou 2018 and Huiliangyou Yuehesimiao had higher TK enzyme activities of 1.18–24.22% and 2.88–20.49%, respectively. It can be seen that for most direct-seeded rice, increasing the activity of the TK enzyme is one of the ways to adapt to low light stress.

### 2.3. Effects of Shading on Dry Matter Accumulation in Direct-Seeded Rice

Dry matter accumulation is crucial to grain yield in rice. Except for the dry matter quality of the aboveground parts at the mature stage in 2021, the other indicators were significantly or extremely significantly different among the varieties, shading treatments, and interactions between varieties and shading treatments (Table 1). In 2021, the shading treatment reduced the dry matter mass of the aboveground part of the direct-seeded rice at the full heading stage and the maturity stage by 2.16–18% and 12.24–27.82%, respectively, except for Jiuyou 27, and the reduction was not significant for C Liangyouhuazhan, and the rest reached a significant level. In 2022, the shading treatment significantly reduced the aboveground dry matter mass of Liangyou 0861, Jingliangyou 534, Longliangyou Yuehesimiao, and Liangyou Huazhan by 10.2–21.7% and significantly increased the aboveground dry matter mass of Yangxianyou 919 by 8.73%. Except for the aboveground dry matter quality of Huiliangyou Yuehesimiao at the maturity stage, which was not significantly affected by shading, the aboveground dry matter quality of other varieties at the maturity stage was significantly reduced by 9.87–24.79% under the shading treatment.

### 2.4. Effects of Shading on Direct-Seeded Rice Yield 

As shown in Table 2 and Table 3, the shading treatment after heading significantly reduced the yield of direct-seeding rice. In the two-year experiment, except for the 1000-grain weight in 2022, which was not significantly different under the interaction of variety and shading treatment, the other indices were significantly or extremely significantly different among variety, shading treatment, and the interaction of variety and shading treatment. In the 2021 experiment, the shading treatment significantly reduced the seed setting rate of direct-seeded rice compared with the control treatment. In addition, the effective panicle number of Jiuyou 27 zhan, Huiliangyou Yuehesimiao, and Longliangyou Yuehesimiao, the total number of grains per panicle of C Liangyouhuazhan, and the 1000-grain weight of Jingliangyou 534 and Huiliangyou Yuehesimiao, the other varieties were significantly reduced under the shading treatment. In 2021, the shading treatment reduced the direct-seeded rice yield by 26.10–34.04%. Among them, Yangxianyou 919 had the largest yield reduction, and Liangyouhuazhan had the smallest. In the 2022 experiment, shading treatment significantly reduced the seed setting rate and 1000-grain weight of each tested variety, and the yield was reduced by 27.45–34.11%. The yield reduction in Zhenliangyou 2018 was the lowest, whereas the yield reduction in Jiuyou 27 was the largest. Further analysis showed that the adaptability of direct-seeded rice to shading stress varied with variety and year and based on the response of yield and yield components to low light stress in the two years, Zhenliangyou 2018 had the strongest shade tolerance and the smallest yield reduction.

### 2.5. Correlation Analysis between the Yield and Yield Components of Direct-Seeding Rice 

Correlation analysis between yield and yield components of direct-seeded rice from 2021 to 2022 (Figure 10) showed that the total number of grains per panicle and seed setting rate were significantly positively correlated with yield. The correlation with yield was seed setting rate > total grain number per panicle > effective panicle number > 1000-grain weight, which indicated that post-heading shading mainly reduced the yield by reducing the seed setting rate and total grain number per panicle of direct-seeded rice.

## 3. Discussion

### 3.1. Changes in the Physiological Characteristics of Directly Seeded Rice under Shading Treatment 

The chlorophyll content is a basic parameter used to evaluate the photosynthetic capacity and physiological responses of plants. The chlorophyll in plants needs to bind to the protein as ‘bound chlorophyll’ to play a role [24,25]. Under shading conditions, the relative content of light-harvesting pigment proteins in the photosynthetic units increases, increasing bound chlorophyll while reducing chlorophyll degradation and photooxidation, resulting in an increase in chlorophyll content after shading [26]. The results of this study showed that the contents of chlorophyll a, chlorophyll b, and total chlorophyll in different varieties under shading treatment were not the same but were significantly higher than those in the control (Figure 1, Figure 2 and Figure 3), which was beneficial for improving light energy capture and light energy utilization in rice. Low-light-tolerant rice varieties maintain a high net photosynthetic rate and chlorophyll content under low-light conditions, which is the physiological basis of their resistance [27,28]. Under low light stress, the chlorophyll a/b ratio decreased (Figure 4) because the proportion of scattered light dominated by blue-violet light increased under shading conditions [29]. To improve the utilization and absorption of blue-violet light in scattered light, the increase in chlorophyll b in rice leaves was greater than that of chlorophyll a. NR activity is affected by many factors, and light is an important factor. When light is sufficient, NR activity is enhanced, which promotes the absorption and utilization of nitrate by plants [30]. Under shade conditions, the NR activity of tobacco, rice, and other crops decreases, resulting in an imbalance between carbohydrate anabolism and nitrogen metabolism. The NR activity of direct-seeded rice first increased and then decreased, and the shading treatment significantly reduced the NR activity of direct-seeded rice (Figure 5 and Figure 6). In 2021, the shading treatment reduced the NR activity of each variety of direct-seeded rice by 5.15–43.09% and decreased by 3.98–37.04% by 2022. Weak light alleviates plant stress by enhancing the activity of protective enzymes such as POD [31]. In this study, from the shading treatment for 7 d (S-7) to the shading treatment for 14 d (S-14), the POD activity of each directly seeded rice leaf increased significantly, which is consistent with previous studies. However, the POD activity of some varieties, such as Zhenliangyou 2018, was lower than that of the control after 28 d of shading treatment, which may be related to the excessive MDA content of the variety and excessive damage to the plant membrane system, which affected the enzymatic reaction process of POD. Low-light stress can lead to an imbalance in active oxygen metabolism and membrane lipid peroxidation in plant cells, resulting in damage to cell membranes and metabolic systems [32]. MDA is a membrane lipid peroxidation marker, and its excessive content indicates cell damage. The content increased significantly under shading treatment, and the increase in the content increased with the increase in shading time, and the degree of damage to the membrane system increased (Figure 8). TK activity is one of the limiting factors of the plant photosynthetic rate, and an increase in its activity can improve photosynthesis in plants [33,34]. Under the shading treatment, the activities of Liangyou 0861, 919, Yuehesimiao, and Liangyou Huazhan increased (Figure 9). The other varieties had a decrease in TK activity during the shading treatment, which may be due to the response of direct-seeded rice to low light stress due to the different characteristics of the varieties.

### 3.2. Changes in Yield under the Shading Treatment

Under low-light stress, plant dry matter accumulation decreases [35]. In 2021, the shading treatment reduced the aboveground dry matter mass of direct-seeded rice at the full heading and maturity stage by 2.16–18% and 12.24–27.82%, respectively, except for Jiuyou 27. In 2022, the shading treatment significantly reduced the aboveground dry matter mass of Liangyou 0861, Jingliangyou 534, Longliangyou Yuehesimiao, and Liangyou Huazhan by 10.2–21.7%. Except for the aboveground dry matter quality of Huiliangyou Yuehesimiao at the maturity stage, which was not significantly affected by shading, the aboveground dry matter quality of other varieties at the maturity stage was significantly reduced by 9.87–24.79% under the shading treatment. Shading greatly reduces crop yield, and the main reason for this reduction is the reduction in grain number and grain weight [36]. The two-year shading experiment showed that the shading treatment significantly reduced the seed setting rate of direct seeding rice, which was 12.15–21.23% and 14.42–26.45% lower than that of the control, respectively. Under low light stress, rice flowering is delayed, the number of empty grains increases, and the seed setting rate decreases [37]. The correlation analysis demonstrated that rice yield in 2021–2022 was significantly positively correlated with seed setting rate and total grain number per panicle (*p* < 0.01) and significantly positively correlated with effective panicle number (*p* < 0.05). The main reasons for the yield reduction of direct-seeded rice in this study were the total grain number per panicle and the seed setting rate. Under the shading treatment in 2021, the maximum yield reduction of Huiyangxianyou 919 was 34.04%, and the minimum yield reduction of Liangyouhuazhan was 26.10%. In 2022, the minimum yield reduction in Shenzhen Liangyou 2018 was 27.45%, and the maximum yield reduction in Jiuyou 27 was 34.11%.

## 4. Materials and Methods

### 4.1. Test Materials and Sites

From 2021 to 2022, eight hybrid rice varieties with similar growth periods and suitable for direct seeding in Sichuan Province were selected as experimental materials.

Table 4 lists the specific test combinations. 

The experiment was carried out in the experimental base of Southwest University of Science and Technology in Fucheng District, Mianyang City, Sichuan Province (31°32′ N, 104°41′ E) in 2021 and 2022. The soil in the experimental planting field was a typical fluvo-aquic soil, and the soil fertility distribution was uniform. The total nitrogen content of the soil in the field is 1.48 g/kg, the total phosphorus is 0.63 g/kg, the total potassium is 15.60 g/kg, the alkali-hydrolyzed nitrogen is 194.97 mg/kg, the available phosphorus is 18.58 mg/kg, the available potassium is 134.24 mg/kg, and the pH value is 7.08. Figure 11 shows the meteorological data collected during the test period (data from the Mianyang meteorological station in Sichuan Province).

### 4.2. Experimental Design 

A randomized block design was used for the experiments. The rice was sown in late May with a plant-row spacing of 33 cm × 16.6 cm, and this was repeated three times. At the heading stage, a layer of white cotton gauze with a diameter of approximately 0.5 mm was hung on a shelf approximately 2.0 m high (50% shading rate), and the test varieties were shaded according to the heading stage. The shading net is 1.5 m away from the canopy, and the surrounding drooping shading net is 0.2 m away from the canopy, which ensures ventilation and air permeability in the shading net and reduces its influence on temperature and humidity. No shading was used as a control. The test field was converted to a special compound fertilizer for rice, according to 150 kg of pure nitrogen per hectare, N:P:K = 26:9:9, base fertilizer:tiller fertilizer:panicle fertilizer = 3:3:4. Other cultivation and management methods were performed per local high-yield cultivation methods. 

### 4.3. Determination of Content and Methods 

#### 4.3.1. Determination of Dry Matter Content 

Three plant samples were harvested at the full heading and maturity stages. The stems, leaves, and ears were packaged. After 30 min of fixation at 105 °C, the samples were dried at 70 °C until the mass was constant, and the dry matter mass of the aboveground part was weighed. 

#### 4.3.2. Leaf Physiological Indices 

Whole flag leaves were collected every 7 d from heading to 28 d to measure the chlorophyll, NR, POD, MDA, and TK contents. 

NR: the unit was U·g^−1^, which was determined by the NR kit of Beijing Solarbio Science & Technology Co., Ltd. (Beijing, China); TK:TK was determined by Shanghai Ruifan Biological Co., Ltd. (Shanghai, China) Plant TK kit, the unit was U·mL^−1^; malondialdehyde (MDA): The MDA kit was determined by Solarbio’s MDA kit, and the unit was nmol·g^−1^; POD:POD was determined by Solarbio’s POD kit, and the unit was U·g^−1^.

#### 4.3.3. Rice Yield and Yield Components 

At the mature stage, five representative rice holes were selected according to the average number of effective panicles for each treatment. After collection, the rice panicles were placed in nylon net bags and dried naturally until the water content reached approximately 13.5%. The number of grains per panicle, number of filled grains per panicle, and 1000-grain weight were measured, and the theoretical yield was calculated. 

#### 4.3.4. Data Analysis 

Analyses of variance (ANOVA) and least significant difference (LSD) tests were used to compare data using SPSS v23 (Chinese version v22.0.0.0) (Statistical Product and Service Solutions Inc., Chicago, IL, USA), with a significance threshold of *p* < 0.05. Figures were constructed using Origin Pro 2023 (OriginLab, Northampton, MA, USA). 

## 5. Conclusions

This study examined the effects of low-light stress on rice varieties in the Si-11 Chuan Basin, a low-light rice-producing region. It found that shading increased chlorophyll content but decreased yield, with Zhenliangyou 2018 and Jingliangyou 534 demonstrating greater tolerance to weak light stress and maintaining higher yields. These findings suggest these two varieties are suitable for cultivation in low-light areas like Sichuan. 

## Figures and Tables

**Figure 1 plants-12-04077-f001:**
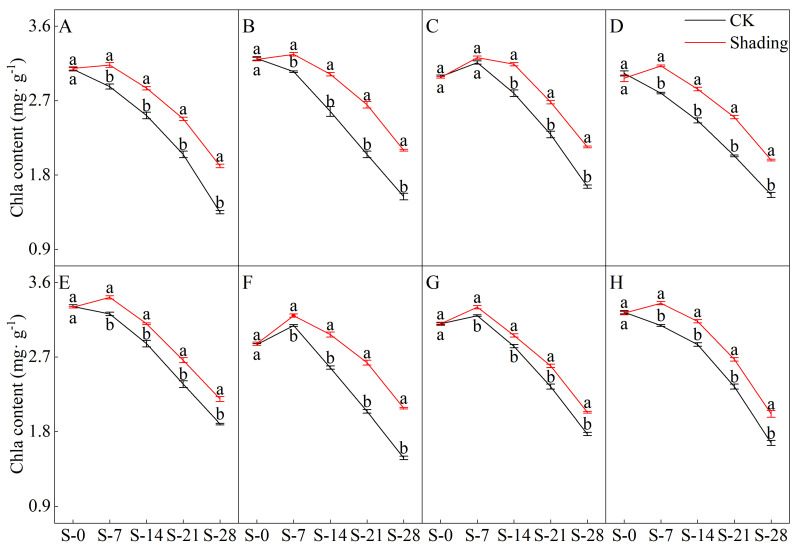
Effects of shading on the chlorophyll a content of direct-seeded rice. Note: (**A**–**H**) represents Zhenliangyou 2018, C Liangyou 0861, Jingliangyou 534, Yangxianyou 919, Jiuyou 27 zhan, Huiliangyou Yuehesimiao, Longliangyou Yuehesimiao, and C Liangyou Huazhan, respectively. S-0, S-7, S-14, S-21, and S-28 represent 0 d of shading treatment, 7 d of shading treatment, 14 d of shading treatment, 21 d of shading treatment, and 28 d of shading treatment, respectively. Different letters are significantly different according to LSD (0.05).

**Figure 2 plants-12-04077-f002:**
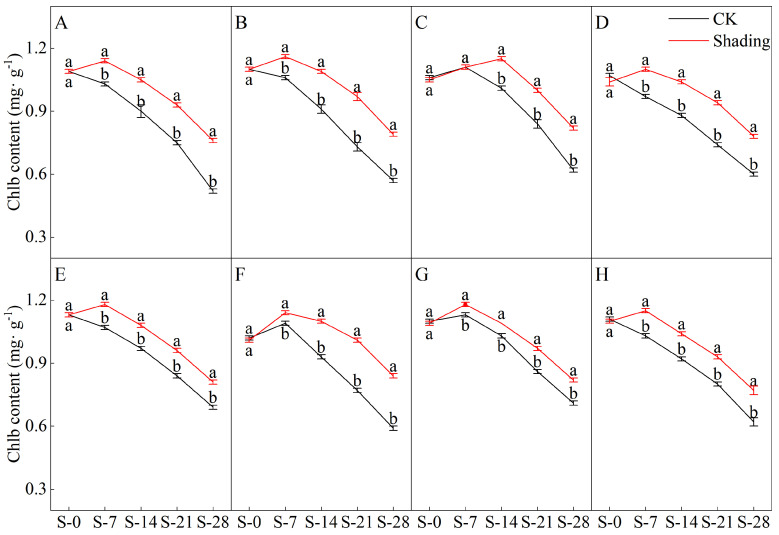
Effects of shading on the chlorophyll b content of direct-seeded rice. Note: (**A**–**H**) represents Zhenliangyou 2018, C Liangyou 0861, Jingliangyou 534, Yangxianyou 919, Jiuyou 27 zhan, Huiliangyou Yuehesimiao, Longliangyou Yuehesimiao, and C Liangyou Huazhan, respectively. S-0, S-7, S-14, S-21, and S-28 represent 0 d of shading treatment, 7 d of shading treatment, 14 d of shading treatment, 21 d of shading treatment, and 28 d of shading treatment, respectively. Different letters are significantly different according to LSD (0.05).

**Figure 3 plants-12-04077-f003:**
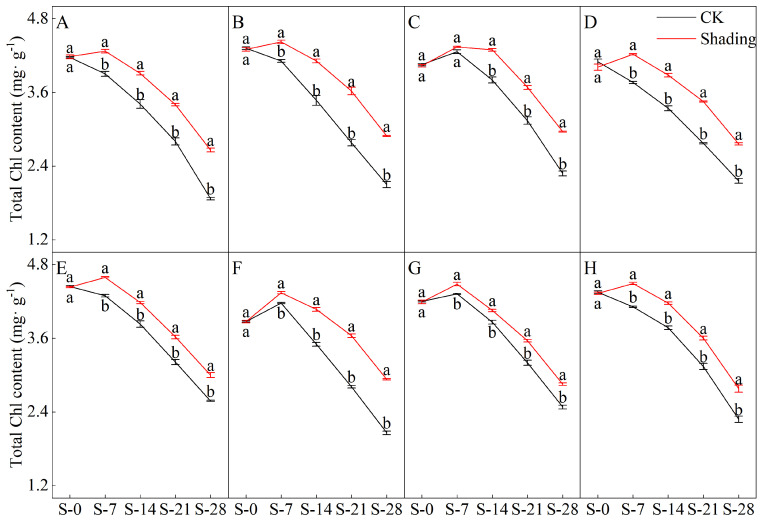
Effects of shading on the total chlorophyll content of direct-seeded rice. Note: (**A**–**H**) represents Zhenliangyou 2018, C Liangyou 0861, Jingliangyou 534, Yangxianyou 919, Jiuyou 27 zhan, Huiliangyou Yuehesimiao, Longliangyou Yuehesimiao, and C Liangyou Huazhan, respectively. S-0, S-7, S-14, S-21, and S-28 represent 0 d of shading treatment, 7 d of shading treatment, 14 d of shading treatment, 21 d of shading treatment, and 28 d of shading treatment, respectively. Different letters are significantly different according to LSD (0.05).

**Figure 4 plants-12-04077-f004:**
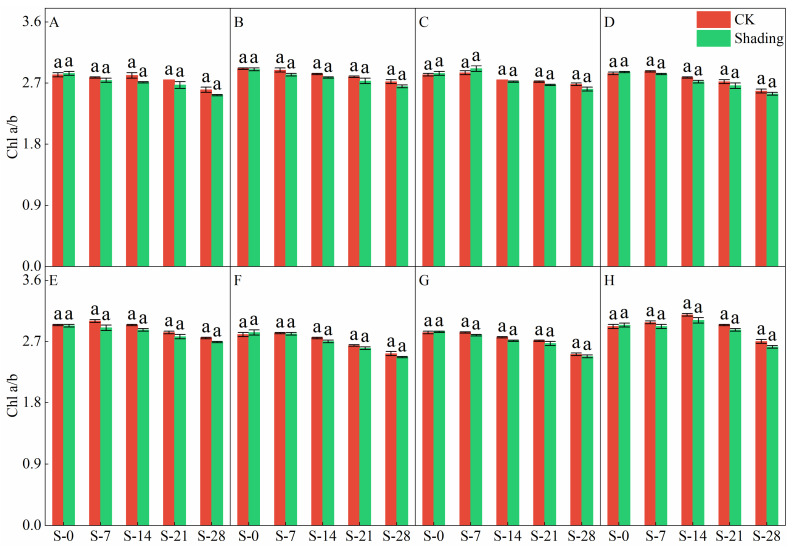
Effects of shading on chlorophyll a/b of direct-seeding rice. Note: (**A**–**H**) represents Zhenliangyou 2018, C Liangyou 0861, Jingliangyou 534, Yangxianyou 919, Jiuyou 27 zhan, Huiliangyou Yuehesimiao, Longliangyou Yuehesimiao, and C Liangyou Huazhan, respectively. S-0, S-7, S-14, S-21, and S-28 represent 0 d of shading treatment, 7 d of shading treatment, 14 d of shading treatment, 21 d of shading treatment, and 28 d of shading treatment, respectively. Different letters are significantly different according to LSD (0.05).

**Figure 5 plants-12-04077-f005:**
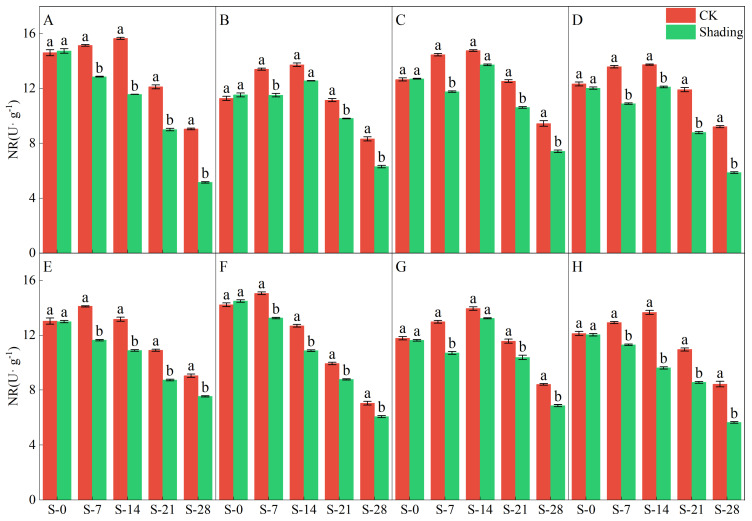
Effects of shading on the NR enzyme activity of direct-seeded rice in 2021. Note: (**A**–**H**) represents Zhenliangyou 2018, C Liangyou 0861, Jingliangyou 534, Yangxianyou 919, Jiuyou 27 zhan, Huiliangyou Yuehesimiao, Longliangyou Yuehesimiao, and C Liangyou Huazhan, respectively. S-0, S-7, S-14, S-21, and S-28 represent 0 d of shading treatment, 7 d of shading treatment, 14 d of shading treatment, 21 d of shading treatment, and 28 d of shading treatment, respectively. Different letters are significantly different according to LSD (0.05).

**Figure 6 plants-12-04077-f006:**
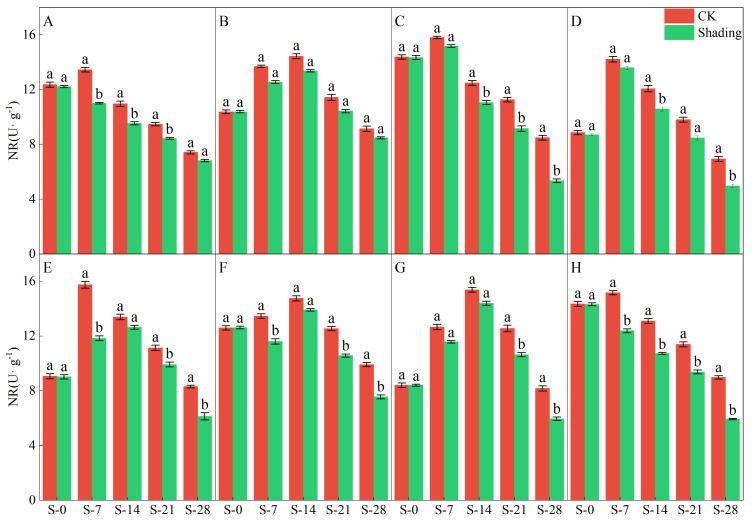
Effects of shading on the NR enzyme activity of direct-seeded rice in 2022. Note: (**A**–**H**) represents Zhenliangyou 2018, C Liangyou 0861, Jingliangyou 534, Yangxianyou 919, Jiuyou 27 zhan, Huiliangyou Yuehesimiao, Longliangyou Yuehesimiao, and C Liangyou Huazhan, respectively. S-0, S-7, S-14, S-21, and S-28 represent 0 d of shading treatment, 7 d of shading treatment, 14 d of shading treatment, 21 d of shading treatment, and 28 d of shading treatment, respectively. Different letters are significantly different according to LSD (0.05).

**Figure 7 plants-12-04077-f007:**
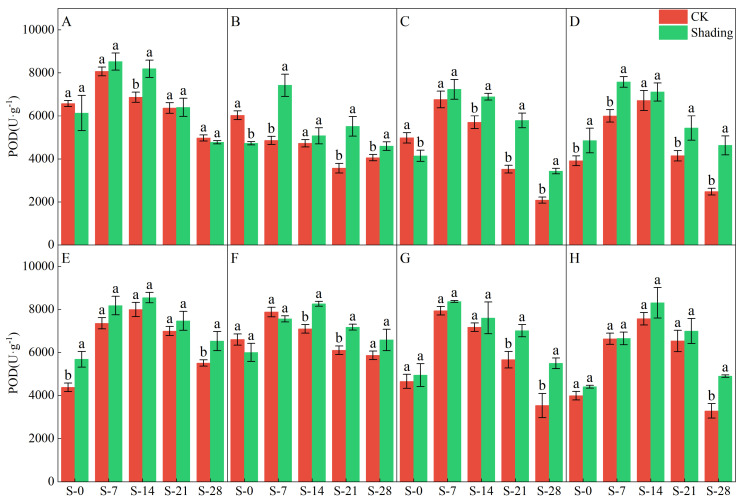
Effects of shading on the POD activity of direct-seeded rice. Note: (**A**–**H**) represents Zhenliangyou 2018, C Liangyou 0861, Jingliangyou 534, Yangxianyou 919, Jiuyou 27 zhan, Huiliangyou Yuehesimiao, Longliangyou Yuehesimiao, and C Liangyou Huazhan, respectively. S-0, S-7, S-14, S-21, and S-28 represent 0 d of shading treatment, 7 d of shading treatment, 14 d of shading treatment, 21 d of shading treatment, and 28 d of shading treatment, respectively. Different letters are significantly different according to LSD (0.05).

**Figure 8 plants-12-04077-f008:**
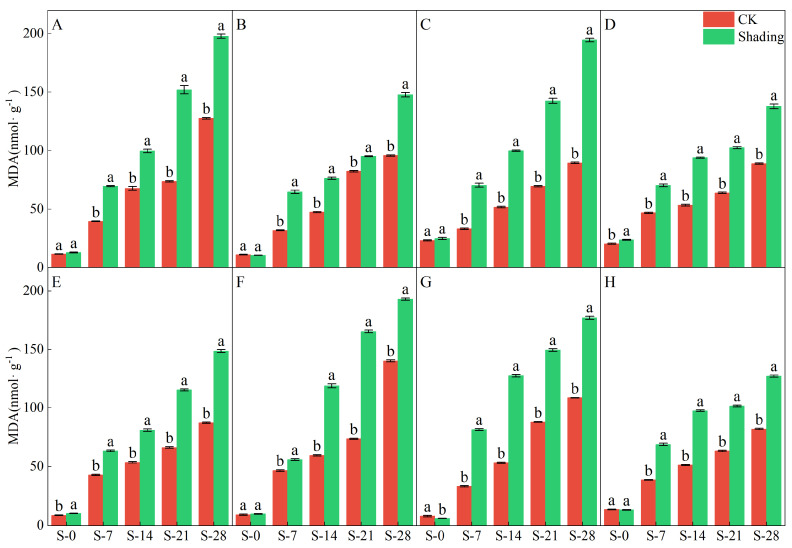
Effects of shading on the MDA content of direct seeding rice. Note: (**A**–**H**) represents Zhenliangyou 2018, C Liangyou 0861, Jingliangyou 534, Yangxianyou 919, Jiuyou 27 zhan, Huiliangyou Yuehesimiao, Longliangyou Yuehesimiao, and C Liangyou Huazhan, respectively. S-0, S-7, S-14, S-21, and S-28 represent 0 d of shading treatment, 7 d of shading treatment, 14 d of shading treatment, 21 d of shading treatment, and 28 d of shading treatment, respectively. Different letters are significantly different according to LSD (0.05).

**Figure 9 plants-12-04077-f009:**
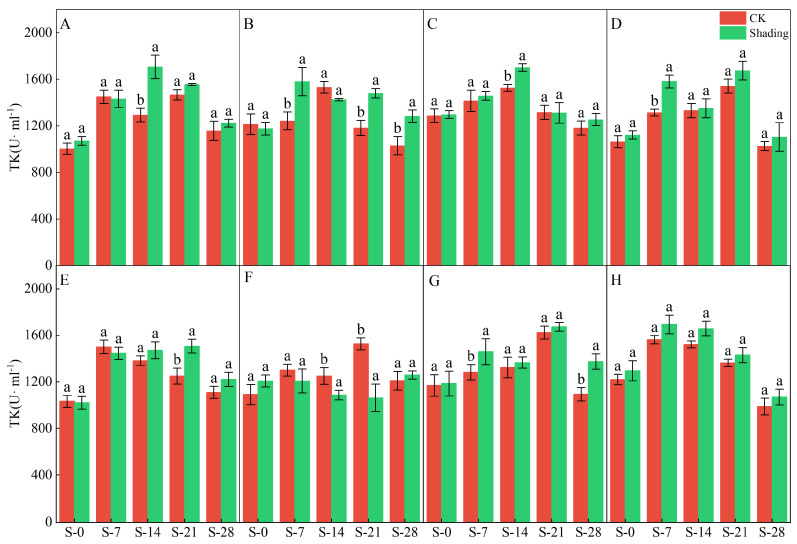
Effects of shading on the TK content of direct seeding rice. Note: (**A**–**H**) represents Zhenliangyou 2018, C Liangyou 0861, Jingliangyou 534, Yangxianyou 919, Jiuyou 27 zhan, Huiliangyou Yuehesimiao, Longliangyou Yuehesimiao, and C Liangyou Huazhan, respectively. S-0, S-7, S-14, S-21, and S-28 represent 0 d of shading treatment, 7 d of shading treatment, 14 d of shading treatment, 21 d of shading treatment, and 28 d of shading treatment, respectively. Different letters are significantly different according to LSD (0.05).

**Figure 10 plants-12-04077-f010:**
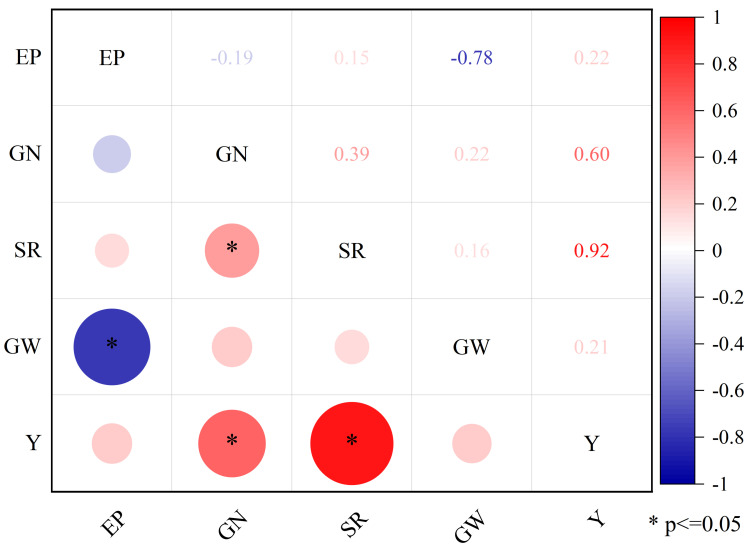
Correlation between yield and yield components of direct-seeded rice in 2021–2022. Note: EP, GN, SR, GW, and Y in the graph represent effective panicle number, grain number per panicle, seed setting rate, 1000-grain weight, and yield, respectively.

**Figure 11 plants-12-04077-f011:**
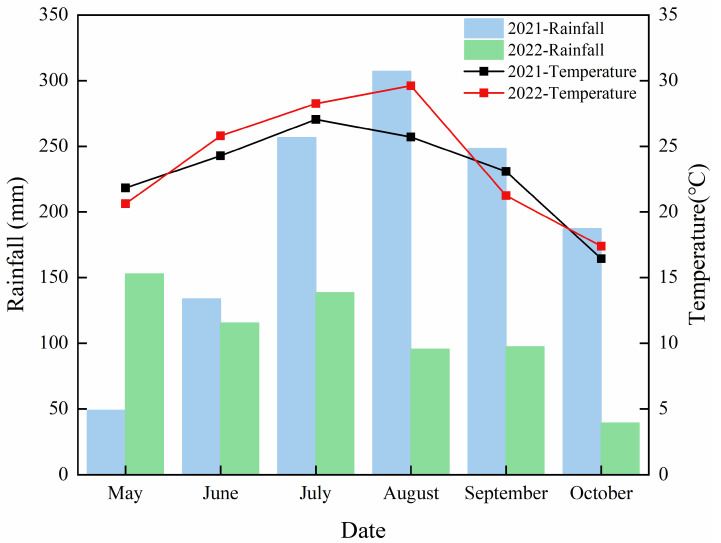
Meteorological data on rainfall and temperature during the test period from 2021 to 2022.

**Table 1 plants-12-04077-t001:** Effects of shading on dry matter mass of direct-seeded rice in 2021 and 2022 (t·hm^−2^).

Variety	Treatment	Aboveground Dry Matter Mass in 2021		Aboveground Dry Matter Mass in 2022	
		Heading Stage	Maturity	Heading Stage	Maturity
Zhenliangyou 2018	CK	11.29 a	16.01 a	11.16 a	16.9 a
	S	10.73 b	13.27 b	11.55 a	15.28 b
C Liangyou 0861	CK	10.64 a	16.19 a	10.75 a	16.09 a
	S	10.22 b	14.12 b	9.65 b	12.54 b
Jingliangyou 534	CK	9.89 a	16.43 a	11.66 a	16.63 a
	S	9.68 b	11.86 b	9.87 b	12.51 b
Yangxianyou 919	CK	11.97 a	16.53 a	11.14 b	18.31 a
	S	10.41 b	13.44 b	12.12 a	15.29 b
Jiuyou 27 zhan	CK	9.92 b	16.42 a	10.28 a	17.58 a
	S	10.56 a	13.90 b	10.08 a	13.90 b
Huiliang Youyuehesimiao	CK	11.95 a	15.83 a	9.77 a	14.84 a
	S	9.80 b	13.72 b	10.46 a	13.82 a
Longliang Youyuehesimiao	CK	11.39 a	16.17 a	12.22 a	16.06 a
	S	10.74 b	13.39 b	9.57 b	12.89 b
C Liangyou Huazhan	CK	10.06 a	14.95 a	10.57 a	16.70 a
	S	9.76 a	13.12 b	9.47 b	13.49 b
F-value	V	42.54 **	2.26	12.55 **	8.19 **
	T	120.92 **	210.50 **	23.65 **	159.09 **
	V × T	26.01 **	2.75 *	13.53 **	2.55 *

Note: CK is no shading; S is 50% shading rate; V, T, and V × T represent the interaction of variety, treatment, and variety and treatment, respectively. Lowercase letters indicate that the dry matter quality of direct-seeded rice is significantly different among the treatments (*p* < 0.05, LSD method). * and ** mean significance at the 0.05 and 0.01 probability levels, respectively.

**Table 2 plants-12-04077-t002:** Effects of shading on yield and yield components of direct-seeded rice in 2021.

Year	Variety	Treatment	Effective Panicles	Grain Number per Panicle	Seeding Rate/%	1000-Grain Weight/g	Yield/(t·hm^−2^)
			/(10^4^·hm^−2^)				
2021	Zhenliangyou 2018	CK	269.81 a	187.37 a	91.56 a	24.83 a	11.49 a
		S	253.33 b	178.94 b	76.18 b	24.13 b	8.33 b
	C Liangyou 0861	CK	218.97 a	191.73 a	86.63 a	26.71 a	9.72 a
		S	207.29 b	165.32 b	76.10 b	25.67 b	6.69 b
	Jingliangyou 534	CK	244.68 a	205.43 a	83.93 a	23.40 a	9.87 a
		S	238.69 b	184.08 b	70.04 b	23.11 a	7.11 b
	Yangxianyou 919	CK	219.76 a	193.97 a	83.75 a	29.89 a	10.67 a
		S	201.78 b	177.16 b	67.45 b	29.19 b	7.04 b
	Jiuyou 27 zhan	CK	246.67 a	195.15 a	87.04 a	27.68 a	11.60 a
		S	244.16 a	180.00 b	71.15 b	26.93 b	8.42 b
	Huiliang Youyuehesimiao	CK	237.39 a	208.35 a	83.39 a	25.27 a	10.42 a
		S	233.22 a	193.53 b	65.69 b	25.04 a	7.42 b
	Longliang Youyuehesimiao	CK	230.22 a	186.17 a	88.20 a	24.68 a	10.09 a
		S	227.10 a	169.60 b	70.11 b	24.22 b	7.08 b
	C Liangyou Huazhan	CK	284.45 a	161.68 a	85.29 a	23.36 a	9.16 a
		S	263.88 b	160.22 a	69.77 b	22.96 b	6.77 b
	F-value	V	367.85 **	238.06 **	6.59 *	830.76 **	19.78 **
		T	161.99 **	750.24 **	336.23 **	116.25 **	538.82 **
		V × T	10.15 **	23.57 **	29.57 **	3.29 *	119.84 **

Note: CK is no shading; S is 50% shading rate; V, T, and V × T represent the interaction of variety, treatment, and variety and treatment, respectively. Lowercase letters indicate that the dry matter quality of direct-seeded rice is significantly different among the treatments (*p* < 0.05, LSD method). * and ** mean significance at the 0.05 and 0.01 probability levels, respectively.

**Table 3 plants-12-04077-t003:** Effects of shading on yield and yield components of direct-seeded rice in 2022.

Year	Variety	Treatment	Effective Panicles	Grain Number per Panicle	Seeding Rate/%	1000-Grain Weight/g	Yield/(t·hm^−2^)
			/(10^4^·hm^−2^)				
2022	Zhenliangyou 2018	CK	255.57 a	187.01 a	87.95 a	24.67 a	10.36 a
		S	249.80 b	185.60 a	67.07 b	24.19 b	7.52 b
	C Liangyou 0861	CK	226.30 a	196.90 a	82.58 a	26.84 a	9.87 a
		S	212.34 b	194.74 a	61.15 b	26.30 b	6.65 b
	Jingliangyou 534	CK	274.14 a	180.49 a	85.83 a	23.06 a	9.79 a
		S	265.91 b	172.43 b	68.22 b	22.57 b	7.06 b
	Yangxianyou 919	CK	194.72 a	193.73 a	90.56 a	30.55 a	10.44 a
		S	182.55 b	174.12 b	72.46 b	29.88 b	6.88 b
	Jiuyou 27 zhan	CK	225.99 a	200.80 a	88.28 a	28.55 a	11.44 a
		S	221.70 a	188.80 b	64.92 b	27.73 b	7.54 b
	Huiliang Youyuehesimiao	CK	227.35 a	199.68 a	88.61 a	25.93 a	10.43 a
		S	221.81 b	183.87 b	66.94 b	25.25 b	6.89 b
	Longliang Youyuehesimiao	CK	211.12 a	206.39 a	89.16 a	24.60 a	9.56 a
		S	210.57 a	187.57 b	71.58 b	23.89 b	6.75 b
	C Liangyou Huazhan	CK	269.05 a	200.55 a	81.35 a	23.43 a	10.28 a
		S	245.01 b	187.11 b	69.63 b	22.93 b	7.32 b
	F-value	V	440.32 **	33.73 **	136.44 **	936.63 **	45.20 **
		T	104.96 **	172.20 **	11,395.86 **	107.96 **	5177.53 **
		V × T	8.13 **	8.09 **	52.31 **	0.61	11.70 **

Note: CK is no shading; S is 50% shading rate; V, T, and V × T represent the interaction of variety, treatment, and variety and treatment, respectively. Lowercase letters indicate that the dry matter quality of direct-seeded rice is significantly different among the treatments (*p* < 0.05, LSD method). ** means significance at the 0.01 probability levels.

**Table 4 plants-12-04077-t004:** Direct-seeded rice varieties in 2021–2022.

Numbering	Variety	Numbering	Variety
1	Zhenliangyou 2018	5	Jiuyou 27 zhan
2	C Liangyou 0861	6	Huiliang Youyuehesimiao
3	Jingliangyou 534	7	Longliang Youyuehesimiao
4	Yangxianyou 919	8	C Langyou Huazhan

## Data Availability

The data presented in this study are available on request from the authors.

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
