# Peer review of "Effects of Low Light after Heading on the Yield of Direct Seeding Rice and Its Physiological Response Mechanism"

_plants, 2023, doi:10.3390/plants12244077_

Round 1

Reviewer 1 Report

Comments and Suggestions for Authors

The manuscript (plants-2754748) submitted by Ma et al. reports a study of shading at heading stage on the performance of yield of rice. There are many such kinds of studies conducted previously. The only difference for this study is the materials used. For this reason, it is important to provide more information about each variety selected for the study, instead of just list them as in Table 4. In other word, why did you select these eight varieties? In addition, the description of the Results 2.1. Effects of Shading on the Chlorophyll Content of Direct-Seeded Rice is inaccurate. Starting from the heading date, the flag leaves gradually begin the senescence process, therefore the chlorophyll contents start to decrease, even in the control plants. Under shading treatment, the chlorophyll contents are not increased, but are just delayed in the decrease trend. For example, in most cases, the values of chlorophyll a, b, or total in shading treatment at the point S-14 are the same as the values of the control at the point S-7, and the same is true for S-21 in shading treatment as S-14 in the control. Furthermore, there are several antioxidant enzymes, why the authors only chose POD? Do other enzymes have the same changing pattern as POD?

Line 138-139. The description of “under the shading treatment, the NR enzyme activity of each variety first increased and then decreased.” Is inaccurate. The varieties in panel A, E, F, and H have the NR activity decreased continuously. Why does the NR activity in the same variety have different changing pattern in 2021 and 2022? Any expanations?

Minor comments:

Table 1. change “quality” to “mass”

Line 385, there is a typo: Solaybio should be Solarbio

Line 393, “number of grains per panicle” repeated.

Comments on the Quality of English Language

English language is good enough, just a few minor typos.

Author Response

Point 1:  The only difference for this study is the materials used. For this reason, it is important to provide more information about each variety selected for the study, instead of just list them as in Table 4. In other word, why did you select these eight varieties?

Response 1: Thank you for your comment. We have already explained clearly(line381-382).

Point 2:  The description of the Results 2.1. Effects of Shading on the Chlorophyll Content of Direct-Seeded Rice is inaccurate.

Response 2:  Thank you for your comment.  We have revised it(line82-94)

Point 3: There are several antioxidant enzymes, why the authors only chose POD?

Response 3:  Thank you for your comment. The reason why we choose pod enzyme is that it can better reflect the changes of rice under low light stress conditions.

Point 4: Do other enzymes have the same changing pattern as POD?

Response 4: Thank you for your comment. Most studies have shown that changes in other antioxidant enzymes under stress conditions are similar to pod enzymes.

Point 5:  The description of “under the shading treatment, the NR enzyme activity of each variety first increased and then decreased.” Is inaccurate. The varieties in panel A, E, F, and H have the NR activity decreased continuously.

Response 5: Thank you for your comment. We have revised it(line 143-169).

Point 6:  Why does the NR activity in the same variety have different changing pattern in 2021 and 2022? Any expanations?

Response 6: Thank you for your comment. The NR enzyme activity of the same variety has different changing pattern in 2021 and 2022, which may be caused by the two-year climate difference in the test site.

Point 7: Table 1. change “quality” to “mass”.

Response 7: Thank you for your comment.We have revised it(Table 1).

Point 8: 1.Line 385, there is a typo: Solaybio should be SolarbioLine 393, “number of grains per panicle” repeated.

Response 8: Thank you for your comment.We have revised it(line 419, 427).

Reviewer 2 Report

Comments and Suggestions for Authors

This study examined the effects of low-light stress on rice varieties in the Si-11 Chuan Basin, a low-light rice-producing region. It found that shading increased chlorophyll content but decreased yield, with Zhenliangyou 2018 and Jingliangyou 534 demonstrating greater tolerance to weak light stress and maintaining higher yields. These findings suggest these two varieties are suitable for cultivation in low-light areas like Sichuan.

Additionally, Figure 1 and Figure 2 lack statistical analysis and require T-tests for validation. For Figure 5, the specific statistical analysis conducted should be mentioned. It is essential to include detailed explanations of the statistical analysis in the figure footnotes to enhance result interpretation and clarity.ed to mentioned which statistical analysis was conducted.

Most notably, the figures often lack descriptions of the statistical analyses performed in their footnotes. Including this information is crucial to provide clarity and context for interpreting the results.

In statistical analysis, the authors mentioned T-test for two group comparison.

Line 281. the protein as 'bound chlorophyll' to play a role [24,25] . -> remove space before period.

Line 288, rice . --> remove space before period.

Author Response

Point 1: This study examined the effects of low-light stress on rice varieties in the Si-11 Chuan Basin, a low-light rice-producing region. It found that shading increased chlorophyll content but decreased yield, with Zhenliangyou 2018 and Jingliangyou 534 demonstrating greater tolerance to weak light stress and maintaining higher yields. These findings suggest these two varieties are suitable for cultivation in low-light areas like Sichuan.

Response 1: Thank you for your comment. We have revised it(line435-440).

Point 2: Additionally, Figure 1 and Figure 2 lack statistical analysis and require T-tests for validation. For Figure 5, the specific statistical analysis conducted should be mentioned. It is essential to include detailed explanations of the statistical analysis in the figure footnotes to enhance result interpretation and clarity.ed to mentioned which statistical analysis was conducted.

Response 2: Thank you for your comment.We have revised  it.(Figure 1 and Figure 2, line116-117,123-124,131-132,138-139,166,173,189,213,223)

Point 3: Line 281. the protein as 'bound chlorophyll' to play a role [24,25] . -> remove space before period.

Response 3: Thank you for your comment. We have revised it(line303).

Point 4:Line 288, rice . --> remove space before period.

Response 4: Thank you for your comment.We have revised it(line310).

Round 2

Reviewer 1 Report

Comments and Suggestions for Authors

In the revised version of the manuscript, the authors addressed most of my comments raised from the previous version. Therefore, I an happy to endorse the acceptance for publication.